# Aptamers as Delivery Agents of siRNA and Chimeric Formulations for the Treatment of Cancer

**DOI:** 10.3390/pharmaceutics11120684

**Published:** 2019-12-16

**Authors:** Ana Paula Dinis Ano Bom, Patrícia Cristina da Costa Neves, Carlos Eduardo Bonacossa de Almeida, Dilson Silva, Sotiris Missailidis

**Affiliations:** 1Instituto de Tecnologia em Imunobiológicos (Bio-Manguinhos), Fundação Oswaldo Cruz. Av. Brasil, 4365-Manguinhos, Rio de Janeiro/RJ CEP 21040-900, Brazil; adinis@bio.fiocruz.br (A.P.D.A.B.); Pcristina@bio.fiocruz.br (P.C.d.C.N.); dilson.silva@bio.fiocruz.br (D.S.); 2Laboratório de Radiobiologia, Divisão de Física Médica, Instituto de Radioproteção e Dosimetria, Comissão Nacional de Energia Nuclear. Av. Salvador Allende S/N., Rio de Janeiro/RJ CEP 22783-127, Brazil; ce.bonacossa@gmail.com

**Keywords:** siRNA, aptamers, cancer, nanoparticles

## Abstract

Both aptamers and siRNA technologies have now reached maturity, and both have been validated with a product in the market. However, although pegaptanib reached the market some time ago, there has been a slow process for new aptamers to follow. Today, some 40 aptamers are in the market, but many in combination with siRNAs, in the form of specific delivery agents. This combination offers the potential to explore the high affinity and specificity of aptamers, the silencing power of siRNA, and, at times, the cytotoxicity of chemotherapy molecules in powerful combinations that promise to delivery new and potent therapies. In this review, we report new developments in the field, following up from our previous work, more specifically on the use of aptamers as delivery agents of siRNA in nanoparticle formulations, alone or in combination with chemotherapy, for the treatment of cancer.

## 1. Introduction

Small interfering RNA siRNA suppress expression of genes by targeting the mRNA expression. Targeted delivery of siRNA to specific cells is highly desirable for safe and efficient RNAi-based therapeutics [1]. However, the half-life of nucleic acids in the bloodstream is short due to the degradation by endo or exonucleases and rapid clearance [2]. One strategy to solve this challenge is developing siRNA delivery systems. Nanoparticles can be defined as particles less than 100 nm in diameter, these systems can be composed by different materials and are employed according to their purpose [3,4]. For this area, the most widely used systems are polymeric particles, nanoemulsions, nanocrystals, solid lipid nanoparticles, and liposomes [5]. The organic particles used for drug delivery application are micelles, liposomes, polymers, dendrimers, and nanogels. They have versatile surface building blocks for efficient endocytosis and loading [6]. There are numerous advantages to using nanoparticles: (I) Increased bioavailability, (II) dose proportionality, (III) decreased toxicity, (IV) smaller dosage form, (V) stability of drugs dosage forms, and (VI) increased active agent surface area resulting in a faster dissolution [7]. Ideally, nanoparticles should be stable in circulation to protect and deliver their therapeutic load (drug) into recipient tissue; have good penetration and retention in the target tissue so that drug release occurs within the therapeutic window; and ultimately be organically excreted to avoid long term accumulation toxicity [8]. Approaches to drug targeting and delivery may be facilitated by the enhanced permeability and retention (EPR) effect. This effect occurs due to the large endothelial tissue fenestrations which are characteristic of the rapid growth of tumor blood vessels. Therefore, the nanoparticles passively diffuse through the microenvironment targeting the tumor tissues [8].

Although nanocarrier technology has improved, its lack of target specificity limits its widespread use, to overcome this issue and address the lack of specificity is the generation of functionalized nanoparticles, i.e., second generation nanoparticles [8]. Nanoparticle surface functionalization occurs through the fixation of a ligand that interacts with specific tissue-specific receptors, to optimize the administration of the target, selectively transporting it to the binding site [9]. One of the advantages of taking drugs directly to specific tissues is the ability to use relatively more toxic and efficient drugs with less risk of collateral damage to other body tissues. In the case of cancer, drugs could be targeted at tumors, avoiding the systemic side effects of traditional therapies. The functionalization includes surface conjugation of chemicals or bio molecules, like folic acid, biotin molecules, peptides, antibodies, aptamers, short, single stranded RNA or DNA oligonucleotides, proteins, and oligosaccharides, to enhance the properties and hit the target with high precision [10]. In order to provide targetability, aptamers have been widely used due to (I) their capacity of binding to target proteins with a high affinity and specificity, (II) having already been shown to have antibody-like characteristics, and (III) the fact that they are relatively smaller and less immunogenic. All of these useful properties make aptamers attractive in therapeutic and diagnostic fields [11].

## 2. Aptamer in the Delivery of Therapeutic Nanoparticles Containing siRNA, shRNA, and miRNA

The origin of siRNA nanoparticles targeted delivery through aptamers dates to 1998. Guo et al. treated T cells with an RNA nanoparticle consisting of a dimer of the packaging RNA (pRNA) derived from the DNA-packaging motor of bacteriophage phi29 loaded with a siRNA for survivin mRNA and conjugated with a CD4 specific aptamer [12]. Hu et al. used the same platform to create a nanoparticle containing an siRNA for ICAM 1 conjugated with aptamer FB4 directed against the mouse transferrin receptor. The in vitro results showed a decreased of ICAM-1 expression and blocked the adhesion of monocytes [13].

From this date, many types of platforms and supports were developed so that siRNA could be targeted through aptamers. To target cells using aptamers, Afonin et al. designed multifunctional siRNA nanoparticles aiming at the silence of multiple HIV-1 genes [14]. To show the feasibility of siRNA delivery using nanorings, Li et al. developed a nanorings construct functionalized with J18 RNA aptamers specific for the human epidermal growth factor receptor (EGFR) [15]. The nanoring design can achieve cell-targeting properties, but the internalization and functional effects of siRNAs bound to aptamer-nanorings were not studied [16]. Zhao et al. developed a nanoparticle to treat anaplastic large cell lymphoma (ALCL). The nanocarrier incorporated both anaplastic lymphoma kinase (ALK) siRNA and a CD30 aptamer onto nano-polyethyleneimine-citrate leading to growth arrest and apoptosis in vitro [17]. Powell et al. tested the possibility of conjugating an aptamer A6 which can bind to human epidermal growth factor receptor 2 (HER-2) receptors on breast cancer cells to P-gp siRNA siRNA-containing nanoparticles. The knockdown of P-gp was increased significantly, decreasing a resistance to chemotherapeutics in the cells [18]. To overcome multi-drug resistance (MDR), an MD- specific aptamer-conjugated grapefruit-derived nanovectors (GNVs) was used for co-delivery of siRNA and doxorubicin (DOX), resulting in a potent anti-tumor activity [19].

Stable nucleic acid lipid particles (SNALPs) also used to deliver siRNA through aptamers. In this study, the authors coupled the transferrin receptor aptamer in SNALP loaded with siRNA. The aptamer-targeting enhanced siRNA uptake and target gene knockdown in cells [20]. Another aptamer–liposome–siRNA delivery system was developed by Alshaer et al. In this model, siRNA complexed with protamine and aptamer anti-CD44, a cell surface biomarker overexpressed in many tumors, was used as targeting delivery for the first time [21]. They confirmed the silencing disease-related genes in tumors [18].

To target prostate cancer cells, Kim et al. used shRNAs against the gene BCL-xL and the PSMA (prostate-specific membrane antigens) aptamer (A10–3) conjugated to a polyethyleneimine (PEI)-PEG construction. In order to amplify the therapeutic response and use different drugs in synergy, the authors also included DOX in aptamer-functionalized nanoparticle, and this approach led to the death of the cancer cells, demonstrating the effectiveness of the combination therapy [22]. Likewise, the PSMA aptamer was used to target delivery of miRNA (miR-15a and miR-16–1) identified as tumor suppressor genes in prostate cancer. The nanoparticle based on polyamidoamine and PEG allowed selective killing of prostate cancer cells [23]. Xu et al. used a multifunctional envelope-type nanoparticle platform constituted of two polymers that co-incorporate siRNA. For the nanoparticle functionalization was used *S*,*S*-2-(3-(5-amino-1-carboxypentyl)ureido)pentanedioic acid (ACUPA) ligand that can specifically bind to PSMA. This study reached an efficient gene silencing in prostate cancer tumor cells in vivo [24].

Lv et al. used Polyamidoamine (PAMAM) dendrimers functionalized with EGFR aptamers to co-deliver shRNA to survivin and erlotinib. Survivin shRNA is used to activate apoptosis while erlotinib is a tyrosine kinase inhibitor. This therapeutic approach in association with Chloroquine acted synergistically to reverse erlotinib resistance in EGFR mutation-positive of non-small-cell-lung cancer (NSCLC) [25].

Another strategy is use of siRNA-aptamer chimera to compose the nanoparticle, and the siRNA-aptamer chimera has emerged as a promising approach for efficient delivery of siRNA to specific cell types, owing to its low immunogenicity, ease of chemical synthesis and modification, and the outstanding targeting specificity of the aptamer [26]. However, in some cases, endosome escape has been reported. To avoid this problem, it has been used siRNA-aptamer chimeras in nanocarriers. Bagalkot and Gao developed a new technology for linking siRNA-aptamer chimeras to carrier nanoparticles [27]. In this new technology, the siRNA block should become the anchor point for interaction with cationic nanoparticles for reduced enzymatic degradation and nonspecific interaction with cells and tissues, whereas the aptamer block should stay on the outside with minimized interaction with the nanoparticle surface. This approach opened new opportunities in targeted delivery based on siRNA-aptamer chimeras.

In recent study, to evaluate cell viability and genes expression in metastatic breast cancer cells, Jafari et al. designed chitosan nanoparticles for co-delivery of Docetaxel and insulin-like growth factor receptor 1 (IGF-1R) siRNA. Docetaxel is a cytotoxic anti-cancer drug approved for the treatment of metastatic breast cancer. IGF-1R signaling is important in tumor growth, development, and its metastasis. The MUC1 aptamer was conjugated to a chitosan based nanocarrier. MUC1-aptamers have been used to targeted drug delivery to several types of cancers, including breast cancer cells [28]. Another study in the cancer field shows an efficiency of EGFR aptamer-conjugated liposome loaded with special AT-rich sequence binding protein 1 (SATB1) siRNA. The genome organizer (SATB1) expression decreased in vitro and in vivo studies with a consequent decrease in choriocarcinoma [29]. A novel targeted delivery platform to mediated gene-silencing in cancer cells was described by Ayatollahi et al.; the 10-bromodecanoic acid (10C) and 10C-PEG was loaded with shRNA plasmid for specific knockdown of BCL-xL protein and was conjugated to AS1411 aptamer for targeting to nucleolin on cancer cells [30]. A synergistic cancer cell death was achieved using a (AS1411) nucleolin aptamer-carbon nanotubes (CNTs) as biological carriers containing BCL-xL-specific shRNA and DOX [31].

Recently, a new types of siRNA delivery by aptamers have been demonstrated. Xu et al. used the three-way-junction (3WJ) RNA to couple epithelial cell adhesion molecule (EpCAM) aptamers aiming to deliver Delta-5-Desaturase (D5D) siRNA. This enzyme acts on dihomo-γ-linolenic acid DGLA, preventing the production of 8-hydroxyloctanoic acid (8-HOA), which is important for inhibiting histone deacetylation in cancer [32]. Li et al. combined pRNA dimers with siRNA and aptamer, and they demonstrated that pRNA, which is a component of a phi29 bacteriophage, can be useful for efficient siRNA delivery. The pRNA-siRNA-aptamer was encapsulated in folate and PEG functionalized chitosan nanoparticles. In that study, the aptamer used was FB4 for a transferrin domain, and the siRNA was c-myc [33]. Wu et al. demonstrated that functionalized cationic nanobubbles decorated with PSMA aptamers are able to carry and deliver FoxM1 siRNA to prostate cancer positive cells and xenographic tumors [34]. One recent study developed a programmable DNA nanostructure that is considered a potential system for drug delivery for non-small cell lung cancer. DNA Prism nanostructure conjugated with MUC-1 aptamers to deliver Rab26 siRNA presented a great cellular uptake efficiency and anti-tumoral activity [35].

In the field of theranostics, this approach can be useful. Theranostics is defined as nanomedicine that can deliver therapeutics to an intended region, diagnose disease, and follow the progression of disease [36]. Recently, interdisciplinary research in this field, which applies theranostics to diagnose and treat cancer, has explosively progressed. Kim et al. 2017 developed a theranostic liposomal system for carrying diagnostic Quantum Dots (QD) and therapeutic siRNA that were coupled to aptamer molecules against EGFR. The theranostic liposomes were evaluated in terms of cancer-targeted siRNA delivery and QD imaging in vitro and in vivo [11]. The anti-EGFR aptamer was used recently as a theranostic delivery system. The authors used two siRNA to provide a combinatorial anti-cancer treatment using BCL-2 siRNA, which interfere with tumor cell proliferation, and siRNA PKC-ι for inhibition of tumor cell migration. Moreover, these vehicles were able to simultaneously delivered QD provided fluorescence signals in internal organs and tumors. This study shows an efficiently delivered targeting of anti-cancer therapeutic siRNAs, as well as fluorescent QDs to tumor tissues, enabling the use of theranostic delivery systems in the treatment and fluorescence imaging of cancers [37].

In this review, we compiled several studies with different types of nanoparticles; despite different types of nanoparticles, aptamer functionalized nanoparticle loading therapeutic siRNA design can generally be done in two ways: (1) incorporated in the nanoparticle itself or (2) conjugated to the nanoparticle surface.

Over the past few years, the investment in therapeutic oligonucleotides (e.g., aptamers, siRNAs, and antisense technology) is constantly increasing. It is projected that clinical testing of aptamer-targeted drugs will materialize over the next 2–5 years However, there are several challenges in the use of aptamers as nanoparticle drivers, including the aptamer structural design, in vivo stability, immunogenicity, cellular uptake, and endosomal escape, that should be considered [16]. There is the possibility of target specificity, RNAi efficacy, and stability of siRNA-aptamers chimeras being affected by chemical modifications. Therefore, it is crucial to find an optimal alternative for each siRNA aptamers chimera to maximize its therapeutic efficacy [38].

In order to increase extracellular stability, incorporation of protective groups, such as thiol-phosphate, 20-fluoro, 20-amino, and 2′-*O*-methyl, in phosphate or sugar of nucleotides improves nuclease resistance of aptamer-siRNA chimeras [39,40]. Another possibility for improving stability is the aptamers PEG conjugation. Furthermore, it has been shown that PEG formulation reduces immune responses in vivo [41]. The thermal stability of siRNA is another parameter to be considered in gene silencing activity when it is conjugated to aptamers [38]. In this context, the computational modeling has been used to guide RNA aptamer truncations, maintaining their functionality, and preventing and minimizing therapy challenges using siRNA.

Several approaches have been developed to facilitate siRNA uptake and nanoparticle entry into endosomes. One possibility is to perform the conjugation of chimeras to short cell penetration cationic peptides (CPPs) that are important for the efficiency in intracellular transport of a variety of macromolecules [42]. Another possibility is the conjugation of the chimeras with peptides from the transactivator of transcription (TAT)region; Diao et al. demonstrated that the survivin siRNA was delivering specifically and efficiently and suppressed tumor growth in prostate cancer in vivo [43]. Inclusion of endosome-disrupting molecules, including PEI and poly (histidine), as well as fusogenic lipids and pH-sensitive lipids in aptamer-siRNA chimeras, have been useful for enhancing siRNA endosomal evasion [24,44,45]. Table 1 summarizes carriers aptamers cited in this section.

## 3. Cancer Immunotherapy Using Aptamers-siRNA

Immunotherapy has emerged as the best clinical alternative for cancer treatment during the last years. It is mostly based in monoclonal antibodies targeting immunological checkpoint pathways, as well as in genetically-engineered cells that directly attack the tumor, like CAR-T cells. Despite the great success of immunotherapy, it works for 20–40% of the patients, depending on tumor type and disease progression stage [46]. So, other classes of molecules and treatments, such as those based in RNA and aptamers, appeared as good alternatives for cancer immunological treatment.

RNA-based immunotherapy is, in concept, more advantageous than those based in recombinant protein expression platforms. It has lower cost, lower immunogenicity, and better penetration in the tumor tissue, which makes it worth developing. In addition, in the last years, a number of pre-clinical and clinical data proved its efficacy, using both aptamer alone and RNAi conjugates.

Antagonistic and agonistic aptamers, acting similar to monoclonal antibodies, were developed to bind to immunomodulating molecules, such as CTLA-4, PD-1, IL-10 R, and LAG-3. The tetrameric antagonistic aptamer selected for CTLA-4, the first one to be developed, was very effective during melanoma murine model studies in vivo [47]. In the same way, multimeric structures of aptamers were shown to have a better performance for agonistic ligands, bringing them together during the activation. It is the case of OX-40 agonistic aptamer, Pratico et al. constructed a multimerized anti OX-40 aptamer using biotinylation and demonstrated a much better biologic effect than the single aptamer, with increased IFN-© production and proliferation, using peripheral blood mononuclear cells as an in vitro model of T cells’ activation [48].

Some receptors of the immune system have an interesting characteristic: They get internalized after T cell activation as a feedback mechanism of controlling over-stimulation. Taking advantage of this feature, some aptamers have been developed as delivery systems to siRNA. The first one was 4-1BB aptamer used to deliver siRNA to T cells, aiming to disrupt mTOR pathway and attenuate IL-2R signaling in CD8^+^ T cells, which, in turn, leads to CD8^+^ T cells persistence inside the tumor site and its elimination [49]. This same aptamer was used to address siRNA for Smad-4, a protein from TGF-β signaling cascade. TGF-β is a key mediator of immune system suppression during progression. This approach was successful in suppressing the negative effect of TGF-βon infiltrating CD8^+^ cells elicited by vaccination with tumor antigens [50].

Cluster of Differentiation (CD) aptamers also have been shown as good alternatives for RNAi delivery. CD molecules are used traditionally for identification of subpopulations of immune cells during flow cytometry studies and, as well as 4-1BB, are frequently internalized after cells activation. For example, using CD4 aptamers complexed with short hairpin RNAs (shRNA), it was possible to silence ROR©T pathway, the key for T_H_17 cells development, opening a possibility of treatment for a number of conditions, mainly related to autoimmune diseases [51]. In the cancer field, Soldevilla et al. used the previously developed agonistic CD40 aptamer to construct shRNA-aptamer chimeras to inhibit SMG1, a kinase that is essential for nonsense-mediated mRNA decay (NMD) initiation. Mice treated with the CD40 aptamer-shRNA chimera showed higher tumor infiltration of lymphocytes [52,53]. Table 2 summarizes aptamers-siRNAs used for immunotherapy cited in this section.

## 4. Tumor-Targeted Aptamers Complexed Directly to siRNAs and Other Agents for Cancer Therapy

Thanks to the unique features of aptamers, such as stability and tumor penetration, its use as tools for delivery of therapeutic agents, including siRNA, has gained great attention from researchers during the last few years.

The idea of a tailored drug for the treatment of a diseased cell avoiding affecting the surrounding healthy cells is not recent, as recalled by Kruspe and Giangrande [54]. It was initially proposed by Paul Ehrlich about one century ago, when he first defined the “magic bullet” and its inherent concept of cell-surface receptor [55]. The route scientists have threaded to reach the current status has grooved a persistent tentative of diverse therapy strategies, some among them based on the conjugation of aptamer with siRNA.

Aptamers, however, due to their anionic characteristics, do not easily transit across the cell lipid bilayer, requiring adequate strategies to overcome the action of those charges, using diverse adjuvants in different structures based on lipids/liposomes, nanoparticles (as seen above), cationic polymers, and peptides.

Aptamer-siRNA conjugates, or AsiCs as they are frequently referred to, comprise an ever-increasing number of structures since 2006 when the delivery of siRNA by an aptamer was described simultaneously by two groups working with prostate membrane antigen (PSMA).

The first aptamer to be conjugated to an siRNA was to treat prostate cancer. The strategy consisted in addressing the tumor trough PSMA, a protein overexpressed in the surface of prostate cancer cells and tumor vascular endothelium, with a PSMA aptamer linked to PLK-1 and BCL-2 siRNAs. The siRNAs were responsible for disrupting the protection against apoptosis of the tumor cells. The tool was successful in eliminating a xenograft prostate cancer in a murine model [56]. After that, many other groups developed similar strategies. Interesting results were obtained for glioblastoma, the most frequent and aggressive primary brain tumor in adults, with a very poor prognosis. Expression and activation of the signal transducer and activator of transcription-3 (STAT3) has been reported as a key regulator of this kind of tumor, so a chimera aptamer antagonistic for PDGFRbr (a receptor overexpressed in glioblastoma, as well as other tumor types) carrying siRNA for STAT 3 was designed. This chimera was able to reduce the anti-apoptotic factors Poly (ADP-ribose) polymerase PARP and BCL-xL, inducing cell death in two glioblastoma lineages in vitro. Moreover, Gint4.T-STAT3 chimera treatment induced significant reduction of tumor growth rate in comparison to control group, as well reduced pro-tumoral factors in a xenograft mouse model of glioblastoma [57]. Similar results were obtained for HER+ breast cancer using a HER-2 sense and antisense bivalent aptamer conjugated to EGFR siRNA. It was able to be delivered to breast cancer cells overexpressing HER-2 and the EGFR gene was efficiently downregulated in vitro, whilst a xenografted tumor growth was supressed in vivo [58]. Another study from the same group showed the performance of a bivalent HER-2-HER-3 aptamer with an EGFR siRNA between them. It was even better in reducing the expression of all three receptors, in inducing cell cycle arrest and apoptosis in vitro and inhibiting tumor growth with a little of target effect in vivo [59]. Aptamer-siRNA chimeras were also used to overcome multidrug resistance by Jeong et al. They engineered a mucin multiaptamer (E18 units) conjugated to RNAsi for BCL-2, with the intercalation of doxorubicin (DOX) between the double helix of the aptamers. Promising results were obtained using this construct to treat MUC-1 overexpressing MCF-7 breast cancer cells. The complex was efficient in delivering doxorubicin to multi drug resistant MCF-7 cells because of increased endocytosis efficiency due to the cluster effect. BCL-2 RNAsi acted synergistically with DOX, increasing the sensitivity of cells for apoptosis and, in turn, decreasing cell viability [60].

An interesting work has used DOX to inactivate cancer stem cells (CSC) by conjugating this drug to engineered RNA aptamers raised against CSC surface marker epithelial cellular adhesion molecule (EpCAM). A modified DNA-RNA hybrid EpCAM aptamer (10-bp GC at stem region) was loaded with 2,5 molecules of DOX per aptamer and this conjugate was capable to release approximated 89.2% of the intercalated DOX just after endocytosis. The authors proved the release of DOX was dependent on low pH that in turn limited the availability of the drug in systemic circulation and tissue interstitium (physiological pH) to affect sensitive organs. Additionally, the complex Apt-DOX was able to target and deliver DOX to EpCAM positive HT29 colorectal cancer cells in dose and time dependent manner and lead to the accumulation and retention of the drug in the cell nucleus. All these characteristics made this construct very efficient in controlling tumor growth overcoming chemoresistance in both tumorsphere formation assay and immunodeficient mice with LDA xenotransplanted with CSC. The aptamer deliver approach developed by Xiang et al. could in fact improve therapeutic index delivering a sufficient drug dose to the critical subcellular target for enough time to eliminate precisely CSCs [61]. The same group extended the investigation to new approaches to tackle cancer stem cells using EpCAM aptamer. They use the aptamer as a driver to survivin RNAi in colorectal cancer cells, both in vitro as well as in colorectal tumor bearing mice. The combination of the surivin downregulation with the action of 5-fluorouacil cytotoxic chemotherapeutic could increase the lethality of colorectal cancer stem cells, as well as tumor control, in the colorectal cancer xenograph model and, consequently, improvement of animal survival [62].

A proof of concept of an approach using an aptamer-liposomal-DOX to target HER3ECD (epitope to trastuzumab binding epitope antigen) with minimum undesirable toxicity is described by Dou Xi et al. The strategy was to conjugate a single-stranded DNA aptamer against HER3 to liposome to improve targeted delivery with less unspecific action of DOX. The aptamer raised against HER3CD was able to bind to HER3 in MCF-7 ^HER3+^, BT 474^HER3+^ but not in 293T ^HER3−^ cells. This aptamer originates the apt-Lip-DOX conjugate that in fact increased the sustained drug release and was able to increase the growth inhibition of MCF-7 ^HER3+^ and BT 474^HER3+^ cells when compared to liposome-DOX and DOX alone challenge. The Apt-Lip-DOX formulation had many advantages when compared to Lip-DOX treatment. There was a higher uptake and retention by MCF-7 tumors in mice. The survival rate of the animals group treated with the apt-Lip-DOX 60 days after chemotherapy was 40% against 0% of the DOX group attained by 36^th^ day after treatment. It is interesting to notice there was any lethality in the Apt-Lip-DOX treated control group that means a very low toxicity of this construct. Additionally, liver and cardio toxicity in the Apt-Lip-DOX treated group was compared to the control NaCl treated group. Finally, the author have demonstrated that the use of aptamers as a driver to target cells unequivocally reduced the availability of DOX in tissues (reduced biodistribution) other than tumor, reducing significantly the cardio toxicity after chemotherapy when compared to the formulation without aptamer [63].

Prusty et al. (2018) developed a very elegant photo-switchable hybrid-aptameric nanoconstruct that efficiently release DOX at high concentration to HGFR (hepatocyte growth factor receptor–cMet) expressing cells. The HyApNc-DOX nanoconstruct was obtained by the combination of two lipid self-assembly single motifs. The first motif was constituted by a lipid-functionalized aptamer trCLN3 with four lipid-modified dU-phosphoramidite 1 at the 5′end of the aptamer. The so-called trCLN3-L4 retained nanomolar affinity to the cMet receptor in tumor cells despite the modification. The second motif was a lipid functionalized 5′-CG rich hairpin ODN designed to carry DOX by intercalation but with a 2′,6′-dimethylzobenzene (DMAB) photoswitch incorporated in the hairpin nanoscaffold. After UV irradiation, DMAB is responsible for the controlled release of DOX inside cell. A lipid-mediated self-assembly approach to build a multi component supramolecular structure resulted in an improved survival to serum nucleases attack and in a better efficiency in the uptake by the target cell and also a precise subcellular distribution of DOX to the cell nucleus, after endocytosis, when compared to the native aptamer [64].

Another approach using aptamers to deliver DOX was developed to target Glioblastoma multiforme tumor cells. A GMT-3 ssDNA aptamer raised against glioblastoma multiforme A-172 cells demonstrates a high affinity to different glioblastoma cell lines (Kd, ~75 nM). Aptamer GMT-3 charged with DOX (molar ratio 1.2 DOX per aptamer) could selectively bind to and inactivate A-172 glioblastoma cells in comparison to the effect noticed in MCF-7 control cells [65].

A great challenge in the treatment of Glioblastoma using chemotherapeutics is to overcome the blood brain barrier (BBB) and to avoid chemotherapeutic drug deliver to the non-tumor tissues. Luo et al. (2017) developed a nanoconjugate delivery system using AS1411 aptamer as a driver to target nucleolin receptor that is overexpressed in glioblastoma U87 MG cells and neo-vascular endothelial cells in way to deliver chemotherapeutics with more precision. AS 1411-Nucleolin complex undergoes endocytosis inhibiting DNA synthesis inducing apoptosis and also inhibiting angiogenesis, that means it has dual targets. The authors incorporated a more efficient cytotoxic drug to AS1411 synthetizing a poly (l-*c*-glutamyl-glutamine)-paclitaxel nanoconjugate (PGGPTX) to achieve an improvement of aqueous solubility and permanence in plasma. All the results obtained by the group demonstrated that the presence of AS1441 aptamer in the complex AS1411-PGG-PTX improved significantly the U87 MG, as well as neo-vascular endothelial cells uptake and internalization, U87 MG spheroid tumor penetration, and U87 MG cell growth inhibition. The same effect could be demonstrated in vivo with an enhanced accumulation of AS1411-PGG-PTX nanoconjugates in glioblastoma cells, its extended retention time in circulation, and its pronounced penetration in glioblastoma tumors. The authors observed that AS1411-PGG-PTX was able to promote an increase in the survival of the tumor bearing mice and an efficient PCX delivery to glioblastoma tissue with a consequently strong cytotoxic effect [66].

One interesting strategy was used by Balasubramanyam and collaborators in a recently published work, where they designed aptamer-siRNA chimeras with great specificity to cancer expressing EpCAM that can deliver siRNAs against chosen oncogenes. In the case, PLK1, BCL2, and STAT3, three important proteins with high relevance in tumor growth, were selected. They tested various chimeras with positive results in cancer cell lines of breast, lung, head, neck, liver, and retinoblastoma [67].

Li et al. (2017) demonstrated an interesting feature of the aptamers paving the way to the action of neoadjuvant cytotoxic drugs, not necessarily carrying drugs molecules in their own structure. They selected a new ssDNA named HL-1 with high affinity and specificity to Maver-I lymphoma cells (Kd = 70 pmol/L). The aptamer presents a G-quadruplex structure and, once internalized by endocytosis, promoted cell cycle arrest in S-phase in around half of cell population after three days of lymphoma cells treatment. They exploit this effect in a combined treatment of lymphoma cells with a cytotoxic cytarabine. The lethal action of cytarabine is predominant over lymphoma cells in S-phase, and they show that several rounds of aptamer treatment was able to synergistically sensitize these cells to the lethal effect of cytarabine [68].

Pancreatic ductal adenocarcinoma (PDAC) could receive a great benefit from an aptamer target drug deliver approach because it is one with the worst prognosis among all other cancers. The limitation of therapeutic drug treatment dose imposed by the side effects resulted from the non-specificity of the available cancer chemotherapeutics contributes significantly to a poor prognosis. Levy and Kratchmer (2018) proposed a new treatment approach using the Waz aptamer (anti-transferrin receptor–TfR) and E07 aptamer (anti-epidermal growth factor—EGFR), two receptors overexpressed in PDAC that are internalized by clathrin mediated endocytosis. The product of the conjugation of these aptamers with thiol-reactive membrane-permeable MMAE (MC-VC-PAB-MMAE) bearing a valine-citrulline linker or the membrane-impermeable auristatin derivative MMAF (MC-MMAF) was tested in Panc-1, MIA PaCa-2, and BxPC3 PDAC cell lines, with promising results [69].

Finally, the MUC1 aptamers were also used to deliver chitosan nanoparticles containing both docetaxel and siRNAs for a co-delivery/co-treatment approached. The conjugation of the aptamer to the nanoparticle increased specificity and cellular uptake of the nanoconjugate, whereas a significant impact of the combination therapy was confirmed in terms of cell viability. Furthermore, successful silencing was confirmed. Although this was a preliminary study conducted in vitro, it also demonstrates the potential of this type of approach [70]. Table 3 summarizes all works cited in this section.

## 5. Concluding Remarks

Today, there are 42 studies in clinical trials using the term “aptamer”, and 64 completed or ongoing studies using the term “siRNA” for treating medical conditions, including: cancer, HIV, and diseases of the immune system (www.clinicaltrial.gov), which indicates that these two therapeutic agents are still on the frontier of knowledge and are tendency in the global market. In a study of market analysis, siRNA and aptamers are the new drugs that biopharmaceutical industries are investing, and the cancer is the most studied disease using this strategy. The global aptamer market was valued at nearly $1.0 billion USD in 2016 and is expected to expand with a CAGR of 20.0% over the period from 2017 to 2025 to attain the value of $5.0 billion USD by the end of 2025. This growth is attributable to technology advances and the introduction of companies at commercial panel. The main players operating in the global aptamer market are AM Biotech (Houston, TX, USA), Aptamer Group (York, UK), Aptamer Sciences Inc. (Gyeongsangbuk-do, Korea), Aptagen LLC (Jacobus, PA, USA), and Base Pair Biotechnologies (Pearland, TX, USA). In regard to siRNA, the therapeutics market is expected to expand significantly until 2025, since the first product approved by FDA is currently available in the market. Major players operating in the global small interfering RNA (siRNA) therapeutics market are GE Dharmacon Lafayette, USA), OPKO Health, Inc. (Miami, FL, USA), Alnylam Pharmaceuticals (Cambridge, MA, USA), Arrowhead Research Corp (Pasadena, CA, USA), Sanofi Genzyme (Cambridge, MA, USA), Genecon Biotechnologies Co., Ltd. (Baesweiler, Germany), Arbutus Biopharma Corp (Burnaby, Canada), Silence Therapeutics AG (London, UK), and Sylentis S.A. (Madrid, Spain) (www.grandviewresearch.com).

Ever since our last revision of the field in 2017 [71], more than 50 new works have been published and revised in this manuscript, which show the potential of this approach. Furthermore, a number of new works have appeared related to nanoparticle formulations of siRNA and other RNA forms, using different types of nanoparticles, and carried successfully by aptamers, whilst some approaches have favored the use of siRNAs in conjunction with chemotherapy agents, with very promising results. A review of some of these technologies in a graphic form is presented in the figure below (Figure 1).

Harnessing the targeting potential of aptamers, together with the silencing power of siRNAs and a long proved cytotoxic effect of chemotherapeutic agents, and bringing them together into a single therapeutic modality, has shown a great co-operative effect. It offers a therapy delivered directly and specifically to the tumor cell, with important proteins for tumor progression or survival silenced, whilst DNA acting cytotoxic agents act on tumor DNA, causing a specific therapy that has the potential to outmatch other approaches that focus on a single modalities.

## Figures and Tables

**Figure 1 pharmaceutics-11-00684-f001:**
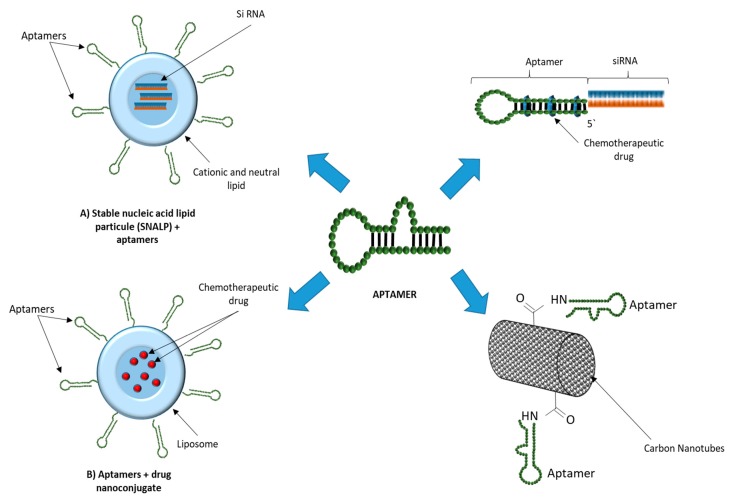
Uses of aptamers in therapeutic delivery.

**Table 1 pharmaceutics-11-00684-t001:** Aptamers for delivery of siRNA.

siRNA/miRNA/shRNA Target Gene	Carrier Aptamer	Efficacy Tests	Ref.
Survivin	CD4	In vitro knock down antiapoptosis factor survivin	[12]
ICAM 1	FB4 (bind to transferrin receptors)	In vitro results showed a decreased of ICAM-1 expression and blocked the adhesion of monocytes	[13]
Human immunodeficiency virus (HIV)-1	epidermal growth factor receptor (EGFR)	In vitro and in vivo gene silencing	[14]
Anaplastic lymphoma kinase (ALK)	CD30	In vitro results of growth arrest and apoptosis	[17]
P-gp	A6 (bind to human epidermal growth factor receptor 2 (HER-2) receptors)	In vitro results about decreasing of resistance to chemotherapeutics	[18]
P-gp	LA1 (specific to multi-drug resistance (MDR))	In vitro and in vivo anti-tumor activity	[19]
GFP	Transferrin	In vitro enhanced siRNA uptake and target gene knockdown	[20]
Luciferase	CD44	In vitro and in vivo results of targeted gene silencing in CD44-positive breast cancer	[21]
P-gp	HER2	In vitro confirmation of silencing disease-related genes in tumors	[18]
BCL-xL	Prostate-specific membrane antigen (PSMA)	In vitro anti-tumor activity	[22]
Genes in prostate cancer	PSMA	In vitro anticancer effect	[23]
Prohibitin 1	PSMA	In vivo gene silencing in prostate cancer tumor cells	[24]
Survivin	EGFR	In vitro and in vivo antitumor effects	[25]
GFP	PSMA	In vitro GFP silencing/Fluorescence imaging (quantum dots)	[27]
Insulin-like growth factor receptor 1 (IGF-1R)	MUC-1	In vitro augment the targeting of pathways involved in tumorigenesis and metastasis	[28]
Special AT-rich sequence binding protein 1 (SATB1)	EGFR	In vitro and in vivo studies of gene expression decreased	[29]
BCL-xL	Nucleolin	in vitro gene silencing and apoptosis	[30]
BCL-xL	Nucelolin	In vitro gene silencing and tumoricidal efficacy	[31]
D5D	EpCAM	In vitro and in vivo inhibition of D5D expression	[32]
c-myc	Transferrin	In vitro and in vivo anti-tumor activity	[33]
FoxM1	PSMA	In vitro and in vivo anti-tumor activity in prostate cancer cells and xenografts in mice	[34]
Rab26	MUC-1	In vitro anti-tumor activity	[35]
bcl-2	EGFR	In vitro interference of tumor cell proliferation	[11]
bcl-2 and PKC-ι	EGFR	In vivo delivery and therapeutic efficacy	[37]

**Table 2 pharmaceutics-11-00684-t002:** Aptamers and siRNA for immunotherapy.

siRNA Target Gene	Carrier Aptamer	Efficacy Tests	Ref.
IL-2 Rα (CD25)	41-BB	In vitro using CHO cells and murine CD8^+^ cells and in vivo using C57BL/6 mice adoptively transferred with OT-I cells and breast tumor 4T1 model	[49]
Smad-4	41-BB	In vitro using murine CD8^+^ cells and in vivo using breast tumor 4T1 model	[50]
RORγT	CD4	In vitro using CD4+ Karpas 299 cells and primary human CD4+ T cells	[51]
SMG-1	CD40 agonist	In vitro using human B- lymphocytes and in vivo using B-cell lymphoma-bearing mice	[52]

**Table 3 pharmaceutics-11-00684-t003:** Aptamers for delivery SiRNA/chemotherapeutics for cancer therapy.

Carrier Aptamers	siRNA or Drug Delivered	Target Cell Lines	Main Effects	Ref.
PDGFRβ (Gint4.T)	STAT-3 siRNA	U87MG and T98G glioblastoma cells	Reduction of anti-apoptotic factors PARP and BCL-xL, inducing cell death in two glioblastoma lineages in vitro. Reduction of tumor growth rate and pro-tumoral factors in a xenograft mouse model of glioblastoma	[57]
HER-2 sense& antisense bivalent	EGFR siRNA	BT474 and SKBR3 breast cancer cells	Downregulation of EGFR in vitro and suppression of xenografted tumor growth in vivo	[58]
bivalent HER-2-HER-3	EGFR siRNA	BT474 and SKBR3 breast cancer cells	Reduction of HER receptors, induction of cell cycle arrest and apoptosis in vitro and inhibition of tumor growth in vivo	[59]
MUC1 multiaptamer	BCL-2 siRNA and Doxorubicin (DOX)	MCF-7 breast cancer cells	More efficiency in delivering doxorubicin. BCL-2 siRNA acted synergistically with DOX increasing the sensitivity of cells for apoptosis	[60]
epithelial cell adhesion molecule (EpCAM)	Doxorubicin (DOX)	EpCAM positive HT29 colorectal cancer cells; SCOV-3 ovarian cancer cell line and T47D breast cancer cell line	Tumor growth control overcoming chemoresistance in both tumorsphere formation assay and in xenotransplanted mice	[61]
EpCAM	Survivin RNAi	EpCAM positive HT29 colorectal cancer cells	The combination of the surivin downregulation with the action of 5-fluorouacil cytotoxic chemotherapeutic could increase the lethality of colorectal cancer stem cells, as well as tumor control, in colorectal cancer xenograph model	[62]
HER-3ECD	Doxorubicin (DOX)	MCF-7 HER3+ and BT 474HER3+ breast cancer cells	Increase in the survival rate in a xenograft mouse model. Reduction of liver and cardio toxicity after chemotherapy when compared to the formulation without aptamer	[63]
cMET	Doxorubicin (DOX)	NCI-H1838 lung cancer cells	Better efficiency in the uptake by the target cell and a precise subcellular distribution of DOX to the cell nucleus, after endocytosis	[64]
Anti A-172 cells aptamer	Doxorubicin (DOX)	Glioblastoma multiforme tumor cells A-172	Selective binding and inactivation of A-172 glioblastoma cells	[65]
Nucleolin (AS1411)	Paclitaxel (PCX)	Glioblastoma U87 MG cells and neo-vascular endothelial cells	Improvement of cell uptake and internalization, spheroid tumor penetration and cell growth inhibition. Enhanced accumulation of AS1411-PGG-PTX nanoconjugates in glioblastoma cells, retention time in circulation penetration in glioblastoma tumors. Increase survival of the tumor bearing mice and an efficient PCX delivery to glioblastoma tissue with consequently strong cytotoxic effect	[66]
EpCAM	PLK1, BCL2, and STAT3 siRNA	NCC RbC 51 retinoblastoma cells; MCF-7 breast cancer cells; Müller Glial Mio M1 cells; PMI 2650 Head and Neck cancer cells and HepG2 Hepatocellular cells	Cell death induction and tumor reduction on RB xenografts tumor model	[67]
HL-1 (anti Maver-I cells)	Cytarabine	Maver-I lymphoma cells	Aptamer treatment was able to sensitize synergistically lymphoma cells in S-phase to lethal effect of cytarabine	[68]
Waz (anti-transferrin receptor—TfR)	Auristatin modified toxins	Panc-1, MIA PaCa-2, and BxPC3 Pancreas Ductal Adenocarcinoma cells	Aptamer conjugates demonstrated to be toxic to cell lines in different extends	[69]
EGFR	Auristatin modified toxins	Panc-1, MIA PaCa-2 and BxPC3 Pancreas Ductal Adenocarcinoma cells	Aptamer-toxin conjugates demonstrated to be toxic to cell lines in different extends	[69]
MUC1	cMET siRNA and Docetaxel	SKBR3 breast cancer cells	Nanoparticle chitosan increased specificity and cellular uptake of the nanoconjugate and successful silencing was confirmed	[70]

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
