# Peer review of "Aptamers as Delivery Agents of siRNA and Chimeric Formulations for the Treatment of Cancer"

_pharmaceutics, 2019, doi:10.3390/pharmaceutics11120684_

Round 1

Reviewer 1 Report

This review entitled “Aptamers as Delivery Agents of siRNA and Chimeric Formulations for the Treatment of Cancer” discuss an important issue. The authors provides the comprehensive information in this manuscript. I have only one suggestion for this manuscript. There was only 1 figure in the current manuscript. To impress your readers, the authors should list the some tables about this content in section 2-4. Tables will be expected to include the targeted genes, species (human or mouse, preclinical or clinical trials), in vivo or in vitro, the type of aptamers, delivery methods, reference, and so on. These tables will help readers quickly summarize the content in the article.

Author Response

Point 1: I have only one suggestion for this manuscript. There was only 1 figure in the current manuscript. To impress your readers, the authors should list the some tables about this content in section 2-4. Tables will be expected to include the targeted genes, species (human or mouse, preclinical or clinical trials), in vivo or in vitro, the type of aptamers, delivery methods, reference, and so on. These tables will help readers quickly summarize the content in the article.

Answer: As suggested, we have included 3 tables in the article, one for each topic (2-4). 

Reviewer 2 Report

The paper “Aptamers as Delivery Agents of siRNA and Chimeric Formulations for the Treatment of Cancer” by Ano Bom et al., is a detailed review which describes the use of aptamers and siRNA technologies in combination for the treatment of cancer, emphasizing the recent developments in the field.

The manuscript is clear and well written with a good selection of the recent works; the topic addressed by this review is actual and very interesting. However some points should be clarified/elucidated in a revised version, in order to make the manuscript more suitable for publication in Pharmaceutics and also more useful to readers.

Below are my comments:

Page 1, line 34-37: in describing the advantages to using nanoparticles for siRNA delivery, the Enhanced Permeability and Retention (EPR) effect should be mentioned by the authors in the introduction paragraph

Page 2, in the paragraph 2: it would be very useful for the readers the inclusion of a brief description about the chemical composition of the cited nanoparticles (containing the nucleic acid) to which the aptamer binds (i.e. if particles have a particular structure, characterized by a lipid or a polymeric nature, other relevant features), as already done by the authors for some of the works mentioned

Page 2, line 83: in the work there is no bibliographic reference concerning the aptamer-liposome-siRNA delivery system developed by Alshaer et al.

Page 3, line 102: please, specify in the text the meaning of NSCLC (non-small-cell lung cancer)

Page 3, line 108: please, change “it have been used” to “it has been used”

Page 4, line 154: please, change “delivered” into “deliver”

Page 5, line 223: in the work there is no bibliographic reference concerning the CD40 shRNA-aptamer chimeras construct developed by Soldevilla et al.

Page 7, line 346: please, change “interest” into “interesting”

Page 7, line 352: please change “has demonstrated” into “have demonstrated”

Author Response

Point 1: Page 1, line 34-37: in describing the advantages to using nanoparticles for siRNA delivery, the Enhanced Permeability and Retention (EPR) effect should be mentioned by the authors in the introduction paragraph

Answer: We have included this item. Please find this paragraph highlighted in red.

Point 2: Page 2, in the paragraph 2: it would be very useful for the readers the inclusion of a brief description about the chemical composition of the cited nanoparticles (containing the nucleic acid) to which the aptamer binds (i.e. if particles have a particular structure, characterized by a lipid or a polymeric nature, other relevant features), as already done by the authors for some of the works mentioned

Answer: We have included this item. Please find this paragraph highlighted in red.

Page 2, line 83: in the work there is no bibliographic reference concerning the aptamer-liposome-siRNA delivery system developed by Alshaer et al.

Answer: We have included this item. Please find this reference highlighted in red.

Page 3, line 102: please, specify in the text the meaning of NSCLC (non-small-cell lung cancer)

Answer: We have included this item. Please find this explanation highlighted in red.

Page 3, line 108: please, change “it have been used” to “it has been used”

Answer: We have done the correction. Please find the correction highlighted in red.

Page 4, line 154: please, change “delivered” into “deliver”

Answer: We have done the correction. Please find the correction highlighted in red.

Page 5, line 223: in the work there is no bibliographic reference concerning the CD40 shRNA-aptamer chimeras construct developed by Soldevilla et al.

Answer: We have included this item. Please find this reference highlighted in red.

Page 7, line 346: please, change “interest” into “interesting”

Answer: We have done the correction. Please find the correction highlighted in red.

Page 7, line 352: please change “has demonstrated” into “have demonstrated”

Answer: We have done the correction. Please find the correction highlighted in red.